# Attitudes, Self-Confidence, and Knowledge of Primary Care Professionals towards School Bullying

**DOI:** 10.3390/healthcare12121230

**Published:** 2024-06-20

**Authors:** María del Carmen Celdrán-Navarro, Ismael Jiménez-Ruiz, César Leal-Costa, James R. Moore, Pedro José López-Barranco

**Affiliations:** 1Servicio Murciano de Salud, 30120 Murcia, Spain; mariacarmen.celdran@um.es (M.d.C.C.-N.); jamesrichard.moore@um.es (J.R.M.); pedrojose.lopez8@um.es (P.J.L.-B.); 2Faculty of Nursing, Campus de Ciencias de la Salud, University of Murcia, El Palmar, 30120 Murcia, Spain; cleal@um.es; 3ENFERAVANZA, Murcia Institute for BioHealth Research (IMIB-Arrixaca), El Palmar, 30120 Murcia, Spain

**Keywords:** bullying, primary healthcare, mixed methods

## Abstract

Attitudes, practices, and knowledge about bullying were evaluated in a sample of 274 primary care professionals, including general practitioners, pediatricians, community, pediatric and school nurses, and residents of these specialties. This study was based on a mixed method with a parallel convergent design without dominance between phases, data were collected concurrently, and conversion of the results from both phases was carried out during data interpretation. The quantitative phase had a cross-sectional observational design, using The Healthcare Provider’s Practices, Attitudes, Self-confidence, and Knowledge Regarding Bullying Questionnaire as an instrument. Descriptive and bivariate analyses were performed, which showed a positive correlation between higher self-confidence and knowledge scores and a greater predisposition to detect cases. However, although the dimensions of attitudes and knowledge yielded generally high data, low self-confidence was evident in addressing this problem. In addition, a lack of clear guidelines in the workplace was expressed, highlighting the need to create and provide specific resources to intervene in bullying in said context, which could develop an improvement in self-confidence, leading to greater well-being for the educational community regarding bullying.

## 1. Introduction

The global scope of the school bullying phenomenon represents a challenge for those responsible for the health and well-being of the educational community. It is defined as aggressive and recurrent conduct intended to cause harm carried out by one or more students against another student and involving a power imbalance. This conduct takes various forms, from physical intimidation to social exclusion [1,2,3]. Its consequences, both immediate and long-term, are potentially devastating, harming physical, mental, social, and emotional health, diminishing quality of life, and can be a cause of suicide [2,3,4,5]. From a physical perspective, recurrent headaches or abdominal pain are common, as well as other gastrointestinal symptoms such as loss of appetite, nausea, or vomiting. Psychosomatic conditions manifest as palpitations, excessive sweating, tremors, and/or dizziness. All these symptoms usually do not follow the common course of the disease or respond to usual treatment. Likewise, alterations in sleep patterns, such as insomnia or nightmares, are observed. Psychologically, individuals often exhibit persistent anxiety and depressive symptoms, including chronic sadness, hopelessness, and low self-esteem, along with irritability and marked emotional fluctuations. All of this can lead to social isolation and significant difficulties in interpersonal and/or family relationships. Regarding academic performance, school bullying is correlated with decreased academic performance, frequent absenteeism, and a decline in motivation and interest in learning, sometimes culminating in premature school dropout, resulting in considerable economic repercussions and limitations in future job opportunities [3,5,6,7,8,9].

The prevalence of school bullying and cyberbullying varies according to the different methodologies applied in studies and the context of each country analyzed. According to various reports, approximately 32% of students worldwide have been victims of school bullying at least once in the past year, with rates ranging from 7% to 74%. Regarding cyberbullying, the global average prevalence is estimated at 17%, with figures varying from 5% to 60% [10,11,12,13].

Current research in Spain reveals that victimization regarding school bullying ranges from 6% to 8.6% [8,14,15]. A recent meta-analysis focused on analyzing the dynamics of school bullying towards individuals with non-normative sexual orientations or gender identities has shown that 77.3% of students participated/identified as bystanders of bullying towards this group and 13.35% identified as aggressors [15]. The Foundation for Children and Adolescents at Risk estimates that in every group of 30 students, there will be 1.8 victims, meaning between 1 and 2 victims of school bullying per classroom [16]. Additionally, with the use of technologies such as the internet and mobile phones, “cyberbullying” emerges. This behavior takes on the characteristics of traditional bullying, being repetitive and deliberate, but it benefits from the advantages provided by online platforms, such as anonymity and the rapid and wide dissemination of information through the network [3,8,17]. Regarding cyberbullying, a research study with a sex distinction, conducted in 2019, revealed that victimization occurs in 15.7% of girls versus 10.7% of boys [17].

When analyzing the role of the aggressor, 89.0% to 99.6% students do not identify with it, which contrasts with the figure of observers identified by 60.4%. These observers indicate that the most repeated behaviors include sending offensive messages that include videos or photos compromising the victim’s integrity in 14.1% of occasions, offensive anonymous calls involving threats and violence in 12.3%, and finally, with 16.3%, stealing social media passwords for identity theft [17]. It is noteworthy that the perpetration of cyberbullying begins in primary education, indicating that boys and girls have access to social networks at very young ages [9,16].

In this context, primary healthcare professionals emerge as fundamental agents in detecting, preventing, and addressing school bullying. Their strategic position gives them direct access to families and the community, providing them with a privileged perspective to identify early indicators of bullying and provide the necessary support to both victims and their closest environments. Additionally, their skills enable them to implement relevant interventions and make referrals to services for specific treatment needs [2,3,5,6,7,8]. Multiple studies highlight the effectiveness of interventions conducted within said context to address school bullying, with both doctors and nurses having implemented successful strategies in consultations and in schools. By offering a holistic approach, these interventions are not limited to identifying cases but also enable comprehensive assessments, allowing for the detection of complex dynamics related to factors such as substance use, the presence of chronic diseases, and other disorders [3,6,18,19]. However, the existing literature on the evaluation of attitudes, practices, and knowledge of these professionals regarding school bullying is limited. It suggests that proactive attitudes and greater self-confidence are crucial factors for detecting and intervening in cases of school bullying. Additionally, notable variability has been observed among professionals regarding their level of knowledge about school bullying, highlighting the need for ongoing training in this area [7,20,21,22].

As a result of the consequences on the comprehensive health of individuals affected by this phenomenon, there has been an increase in primary care consultations [23]. Therefore, it is imperative that these professionals are adequately trained to identify and address these symptoms, providing early and effective intervention to mitigate the negative impact of school bullying on the health and well-being of students [2,3,6,7]. Therefore, based on the hypothesis that better attitudes and knowledge about bullying, combined with good self-confidence, improve the ability to intervene in these cases, we propose the following objective of this study: to analyze the attitudes, knowledge, self-confidence, and training needs of primary care professionals regarding school bullying.

## 2. Materials and Methods

### 2.1. Design

This study was a mixed-methods study with a parallel convergent design without domain status between phases. Data collection was conducted concurrently, and the conversion of the results from both phases was carried during the interpretation of the results [24,25].

For the quantitative phase, a descriptive cross-sectional study was conducted using a self-administered questionnaire. This questionnaire included open-ended questions to illustrate the quantitative results and achieve a broader understanding of the reasons and barriers for screening bullying and the clinical practice developed to address this phenomenon. For the qualitative section, a phenomenological approach focused on the opinions of the respondents was used [26].

### 2.2. Participants and Sample Size

The sample focused on primary care professionals who were likely to be in contact with the population affected by school bullying. Specifically, community, pediatric, and school nurses, as well as general practitioners and pediatric physicians, were included. Additionally, both medical residents and nursing residents from said specialties were included.

A non-probabilistic sampling method was used. The final sample consisted of 274 participants. The characteristics of the sample are detailed in the results section. Of the 274 participants, 94 responded to the open-ended questions, which constituted the qualitative phase of this study. The codes assigned to the respondents inform about their professional category, followed by the order number in which the response was received. EIR/MIR = nursing or medical resident; MFYC = general practitioner; EFYC = family and community nurse; EP = pediatric nurse; MP = pediatric physician; EE = school nurse.

### 2.3. Study Variables and Instruments Used

The instrument used was the validated version of the Healthcare Provider’s Practices, Attitudes, Self-confidence, and Knowledge Regarding Bullying Questionnaire (Table 1) (HCP-PACke) [27]. This is divided into 5 blocks: sociodemographics, attitudes, knowledge, self-confidence, and training needs, of which there are 3 measurable subscales: attitudes, knowledge, and self-confidence. We can define each of them in detail as follows:

Sociodemographics: A section composed of 5 questions that provide information about the gender of the respondent, their profession, professional experience, and specialist qualification and method of obtaining it, and a final one that indicates whether they regularly attend children aged 0–18, serving as a differential screening within the questionnaire itself, providing the option to respond to the attitudes subscale if the answer is positive.

Attitudes: This examines the professionals’ disposition towards attending to victims of bullying within the framework provided by the institution or service. It consists of 6 questions, which can be scored within a range of from 6 to 24 points.

Self-confidence: This evaluates the healthcare providers’ own perceptions regarding their ability to assist individuals affected by bullying. It comprises 8 questions with scores ranging from 8 to 32.

Knowledge: This measures the conceptual level regarding bullying. This part consists of 16 questions, with scores ranging from 16 to 64 points.

These subscales are scored on a four-point Likert scale as follows: strongly agree (4 points), agree (3 points), disagree (2 points), and strongly disagree (1 point). The approximate response time is 10 min. In addition to the dimensions that comprise the mentioned instrument, questions are also asked about training needs and the development of daily clinical practice through a series of closed- and open-ended questions. The open-ended questions used were as follows: ‘In my practice, I assess bullying due to other reasons not listed above. Please share those reasons below’; ‘If there are other procedures you follow when a patient has been bullied, or has bullied others, please share them below’; and ‘I do not assess bullying due to other reasons not listed above. Please share those reasons below’.

### 2.4. Data Collection Process

Data collection took place electronically through a link to an online survey, which was distributed through social networks and instant messaging channels. The data were stored on the UMU surveys platform and were transferred to Excel format for exploitation through the statistical package.

### 2.5. Analysis of Data

Descriptive statistics analyses were carried out to determine the characteristics of the participants and the scores of the scales used. The qualitative variables were expressed as frequencies and percentages. For the quantitative variables, the mean and standard deviation (SD) were calculated. The analyses were carried out with SPSS (Statistical Package for the Social Sciences) v. 24.0. To determine the normal distribution of the sample, the Kolmogorov–Smirnov test was applied. After the normality analyses were carried out, it was decided to perform the non-parametric Mann–Whitney U test to compare the dichotomous variable “Screening for bullying” and the quantitative variables “Attitudes, Self-confidence and Knowledge”. The results were considered statistically significant when the *p* statistic < 0.05 with a 95% confidence interval.

The qualitative methodology used in this study was processed through thematic analysis following the steps described by Braun and Clarke [28]. It began with familiarization through reading and re-reading the data and taking notes of initial ideas. Open coding of the data was conducted inductively, systematically generating emerging codes as the process progressed. Subsequently, these codes were grouped into potential themes, which were reviewed and refined to ensure their coherence and relation to the entire data set, thus creating a thematic “map” of the analysis. Throughout the process, comparisons were made between two researchers who carried out each phase of the analysis separately, seeking agreement after each one, allowing for increased methodological rigor through investigator triangulation and data collection until theoretical saturation was reached. Finally, the most representative verbatim quotes were selected, and a final analysis was conducted to produce a report of the results. The ATLAS.TI.24 software was used for the analysis.

### 2.6. Ethical Considerations

This research was conducted in accordance with the legal and ethical aspects that protect the dignity, autonomy, and benefit of the participating subjects. Therefore, this study was conducted respecting what is imposed by the following current legislation: Organic Law 15/1999, of 13 December, on the Protection of Personal Data/Ley Orgánica 15/1999, de 13 de diciembre, de Protección de Datos de Carácter Personal [29]; Organic Law 3/2018, of 5 December, on the Protection of Personal Data and guarantee of digital rights/Ley Orgánica 3/2018, de 5 de diciembre, de Protección de Datos Personales y garantía de los derechos digitales [30]; and the WMA Declaration of Helsinki—ethical principles for medical research involving human subjects [31].

The participants were duly informed that participation was voluntary, and all respondents gave their informed consent and were warned of the possibility of revoking this and withdrawing the data provided at any time during the course of this research.

The anonymity of the participants was ensured, excluding personal data, thus protecting the right to confidentiality.

In addition, this study was approved by the Ethics and Research Committee of Health Areas II and VIII of the Murcia Health Service (included in the following protocol: EAP AND BULLING 2021-01, 09/28/21).

## 3. Results

### 3.1. Description of Participants

Table 2 shows the participants’ main characteristics. Regarding sex, 81.8% (224) were women and 18.2% (50) were men. In reference to the health profession they were in, they were as follows: general practitioners, 18.2%; pediatricians, 8.4%; community nurses/general practice nurses, 38.7%; pediatric nurses, 7.7%; school nurses, 8.8%; medical residents and nursing residents, 5.1%; and finally, in the others category, we found 14 nurses who declared themselves “generalists”, with this category being 5.1% of the participants. The study participants had an average of 14.1 years of professional experience (SD = 10), with a minimum of 1 year and a maximum of 45 years.

### 3.2. Attitudes, Self-Confidence, and Knowledge of Health Professionals Regarding Bullying and Its Evaluation

In Table 3, we can see the average scores of the items included in the dimensions of attitudes, self-confidence, and knowledge and of the dimensions of the questionnaire. In the dimensions of attitudes towards addressing bullying and knowledge about the phenomenon and its approach, a high score was obtained, while in the dimension of self-confidence, a medium score was obtained.

When the scores of the questionnaire on attitudes, self-confidence, and knowledge about bullying and the screening carried out by health professionals were compared, no significant differences were found between the groups (Mann–Whitney U (95% CI) = 7468.0; *p* = 0.208). However, when comparisons were made between the self-confidence scores and whether or not they had screened for bullying, it was evident that the professionals who had screened for bullying showed higher levels of self-confidence (Mann–Whitney U (95% CI) = 5430.5; *p* = < 0.001). Along the same lines, it was identified that the professionals who did perform the bullying screening obtained higher scores in the knowledge dimension (Mann–Whitney U (95% CI) = 6359.0; *p* = 0.002) (Table 4).

The evaluation of bullying is a critical aspect in the work of health professionals and seems to depend on their knowledge and self-confidence. Through open questions, different reasons were identified that influence the decision of whether or not to evaluate this problem. On the one hand, professionals who chose to evaluate bullying highlighted the importance of their professional competence and their ethical duty. For them, evaluation is an integral part of their responsibility as health professionals: “Because it is one of the competencies of family and community nurses”, EIR/MIR1; “Because it is ethical, moral and all health professionals should do it”, MFYC1; “Because I consider it a health problem, the responsibility of community nurses”, EFYC4. Likewise, they highlighted the importance of screening for bullying due to its high impact on the physical and mental health of young people, and, therefore, it should be treated as a public health problem: “I evaluate it because I think it is important to raise awareness among the little ones of what bullying another child entails. And to prevent it”, EFYC6; “To find out if he has any psychological problem. In order to help the child and seek remedies for bullying. And as an important component of the comprehensive assessment”, EIR/MIR5.

On the other hand, there are barriers that make the evaluation of bullying difficult. Consistent with the bivariate analyses, a lack of specific training and tools or knowledge of specific screening tools was one of the most cited reasons. Many professionals expressed that they do not feel qualified to address this problem due to a lack of knowledge and adequate resources: “I have no training. If they ask me expressly if I pay attention. I think that the teachers are very attentive”, MFYC5; “I do ask about relationships at school, I am in contact with the Health and School nurse in my area, but there are no questionnaires or tools available in the daily consultation or in the medical history. I have not received training either”, EP5; “Because I have not been adequately trained for it, although I did see some cases in child and adolescent mental health during my rotations as a resident”, EFYC34. This lack of training expressed by the participants resulted in a lack of self-confidence to address this problem: “Lack of confidence in my ability to correctly evaluate harassment situations. Avoid misunderstandings or conflict situations with the adolescent and his or her family member or companion”, EFYC10. The professionals also argued that the low demand for consultations related to bullying also plays an important role, since some professionals indicated that they rarely receive cases related to this topic in their consultations: “Except for physical injuries or prolonged nervousness/anxiety, parents They have not consulted me for this reason”, EIR/MIR12; “I ask about how school is going, but not specifically about bullying. If I ask about relationships with colleagues”, EFYC32; “We only do a check-up at age 14 if it has not been done in pediatrics, otherwise they only come for consultation for medical problems and if it falls within the anamnesis, it is addressed and if not”, MFYC11. Furthermore, some health professionals perceived that the issue is dealt with by other professionals or entities, such as schools, which can lead to a perception that it is not their responsibility to address it. “In my opinion it is an issue that is understood to be dealt with in its entirety in schools”, EFYC25; “It must be treated from the school itself with sufficient forcefulness given the lack of authority of the teachers and the provision of school child and adolescent psychologists. The tendency to medicalize it in health centers responds to the deficiencies in this sense”, MP9.

### 3.3. Resources Linked to Addressing and Preventing Bullying in Health Centers

Regarding the resources available in health centers for addressing and preventing bullying, only 12% of the sample stated that they had specific protocols or guides, and 4% had educational materials for said approach (Table 5).

### 3.4. Description of Clinical Practice Linked to the Approach and Prevention of Bullying

Table 6 describes aspects related to the clinical practice developed by primary care professionals for the prevention and addressing of bullying. The majority (47.8%) of the professionals had intervened frequently or very frequently in suspected cases of bullying. They had also used referrals to mental health professionals as a major intervention. These data are supported by the answers given in the qualitative part. These practices involve coordination with other health professionals and with the school team to offer a comprehensive approach to addressing cases of bullying: “Coordination with the psychology department to hold workshops at the educational center”, EE5; “Communication with the health team (pediatrician, school nurse, nursing coordinator)”, PS12; “Contact with the social worker at the health center”, EFYC7. However, referral or contact with school counselors was an underestimated intervention, since 46.7% did not apply this action, although, in the open responses, they talked about the importance of “Involving the school environment to address bullying”, EIR/MIR4. However, the health professionals preferred the school nurse as a link between health and educational centers: “Communication with the school nurse to mediate with the school”, EFYC2, “Communication to the school through the school nurse and referral to the pediatrician”, EFYC4. Finally, and within the clinical practice carried out by professionals, advice to relatives of patients who have suffered bullying stands out; 92.2% of the participants affirmed that they agree with this measure. These data are corroborated in the qualitative testimonies, where the need to involve the family in the process was highlighted, emphasizing communication with parents and the recommendation of concrete actions to address the problem: “Communication with parents and offering educational tools problem management”, MP1; “Recommendation to parents to talk to the student’s tutor and take action”, MP2; “Review of family and school supports, establishment of an action plan”, MFYC3.

## 4. Discussion

In response to the objectives of this research, attitudes, self-confidence, knowledge, resources used, and clinical practice linked to the phenomenon of bullying were analyzed, establishing a positive relationship between self-confidence and knowledge and bullying screening in primary care. Together, the quantitative and qualitative data provide a complete and robust picture of health professionals’ assessment of bullying. While the qualitative testimonies offer an in-depth understanding of perceived motivations and barriers, the quantitative data support these findings by showing how screening performance relates to self-confidence and knowledge of professionals.

Specifically analyzing the dimension of knowledge, it was found that, in general, the respondents had a good conceptual level about bullying. Various studies show that there is a gap in the desired level of knowledge about bullying among primary care doctors and nurses and that this has an impact on its correct approach [20,32,33]. Therefore, the importance of increasing training in this area is highlighted for the better management and detection of these situations in the population, in an interdisciplinary manner, uniting family and pediatric nurses and doctors with school nurses and mental health services. Consequently, creating a support network that connects educational, care, primary, and mental health centers through school nurses can enrich this process [7,20,32,33,34].

In contrast to our results, the data presented by Hutson et al. do not show significant differences between a higher level of knowledge and a predisposition towards the detection and screening of bullying but greater self-confidence and better attitudes in professionals who detected bullying [7]. The same association between the predisposition to detection and the level of knowledge occurred in a sample of doctors specialized in counseling populations with genetic problems [35].

A notable aspect related to self-confidence is the low score obtained in this dimension. This agrees with the findings of other articles that explore self-confidence in both primary care health providers and hospital care who admit to feeling poorly trained in the approach to, the recognition of signs and symptoms, and the actions aimed at resolving a health demand focused on the dynamics of bullying [7,20,35]. As occurred in the testimonies in the present study, various studies attribute the decrease in confidence in addressing this phenomenon to the scarcity of specific professional training opportunities on school bullying [20,32,33,34]. They state that interdisciplinary work and an assignment of standardized care guidelines would improve the safety of professionals and patients in this practice [7,20,34].

As can be seen in the results, the attitudes dimension expressed high scores on average. This agrees with what has been stated in other publications, where the disposition of primary care professionals was positive towards bullying, since they conceived it as a pertinent problem to face [7,20,33]. Condon et al. state that despite a lack of any definition of the specific role in addressing bullying, primary care professionals see their involvement in this practice as a priority [20]. Likewise, Sklar explains that there is receptivity on the part of those involved in assisting with bullying, seeing its screening in the population as viable through interdisciplinary work that helps to improve the monitoring of cases [34].

If we analyze the professional practice carried out by the respondents, they corresponded with Hutson et al., where registration in medical history, referral to mental health, and counseling were the most frequently carried out practices, while contacting the educational center counselor and providing educational material support were the ones that were least carried out [7]. These practices can be aligned with what is stated in two studies that show that the most prevalent practices are aimed at detection, registration, treatment, and referral, but in contrast, they were found with little coordination between health and educational establishments and with few support resources in specific consultations [20,36].

### 4.1. Limitations

We can find a limitation in the response rate of the questionnaire, with a great difference in the number of responses between professional groups, giving rise to underrepresented groups, which prevented a more exhaustive inter-category analysis. On the other hand, since it was a self-administered questionnaire, there may be variability in the understanding of the questions and, thus, a response bias. However, the combination of approaches enriches and complements the understanding of the topic and highlights the importance of addressing training and institutional support needs to improve attention to bullying.

### 4.2. Implications for Practice

In relation to the resources available to address bullying, we can highlight the lack of clear guidelines (protocols or guides) in the workplace [7,20,36]. This coincides with the data expressed by Condon et al., who also state that not having protocols or guides generated uncertainty about the scope of the role of their competencies in this problem [20].

In this sense, despite not having specific documents or manuals for assistance with bullying in classrooms, the importance of asking open questions about friendships and peer groups to reveal bullying situations is highlighted, thereby suggesting the need for guidelines that are more structured to address this issue, these being data that coincide with what was clarified in our research [7,20,36].

All of this requires a greater involvement of clinical practice in the development of pre-established materials and resources to help primary care staff identify, address, and manage bullying in their consultations, as well as implement them through regulated and specific training with continuous updating [7,20,27,34,36].

## 5. Conclusions

The present research reveals a positive association between greater self-confidence and knowledge, which are linked to a greater intentionality in detecting cases of bullying by primary health care professionals. The evaluation of bullying is a complex task that faces various challenges, from a lack of training to the perception of roles and responsibilities. However, it is crucial that health and education professionals work together to address this problem and provide the necessary support to young people who suffer from it.

Although the high scores in their corresponding dimensions showed that there were good attitudes and knowledge in the entire sample studied, their self-confidence was relatively low in terms of their perceived ability to handle those affected by bullying in the classroom. Linked to this, it was observed that the means available to assist the population involved in this phenomenon in health services were perceived by the participants as scarce, or they are unaware of their existence. Linked to this, we can consider that in order to improve self-confidence in addressing the phenomenon of bullying, it is imperative to implement more structured guidelines for addressing it. When analyzing these dimensions, we can perceive an opportunity for improvement that requires the direct involvement of healthcare professionals in the development of specific resources, such as guides and protocols, in cooperation with experts in the educational field. It is essential for these professionals to engage in the creation and use of these documents and to integrate them into their ongoing training. Additionally, it is crucial to promote close interdisciplinary collaboration within the educational context. This will ensure that responses and actions addressing the issue of bullying are more comprehensive, complete, and enriching, thereby benefiting the health and well-being of the entire educational community.

## Figures and Tables

**Table 1 healthcare-12-01230-t001:** Reliability indices for the Healthcare Provider’s Practices, Attitudes, Self-confidence, and Knowledge Regarding Bullying Questionnaire.

HCP-PACke			
Dimensions	Attitudes	Self-Confidence	Knowledge
Items	AC1, AC2, AC3, AC4, AC5, AC6	Auto1, Auto2, Auto3, Auto4, Auto5, Auto6, Auto7, Auto8.	C1, C2, C3, C4, C5, C6, C7, C8, C9, C10, C11, C12, C13, C14, C15, C16
Cronbach’s alpha	0.735	0.940	0.895

**Table 2 healthcare-12-01230-t002:** Descriptive statistics of the sample characteristics.

Variables n = 274				
Gender	n	(%)		
Female	224	(81.8)		
Male	50	(18.2)		
Profession (%)			Gender n (%)	
			Female	Male
General Practitioner	39	(14.2)	30(76.92)	9(23.07)
Pediatrician	23	(8.4)	17(73.91)	6(26.08)
General Practice Nurse/Community Nurse	106	(38.7)	85(80.18)	21(19.81)
Pediatric Nurse	21	(7.7)	19(90.47)	2(9.52)
School Nurse	24	(8.8)	19(79.16)	5(19.23)
Medical and Nursing Resident (EIR/MIR)	47	(17.2)	42(89.36)	5(10.63)
Others *	14	(5.1)	12(85.71)	2(14.28)
Work Experience (years) M (SD)	14.1	(10)		
Perform screenings on bullying n (%)				
Yes	90	(32.8)	73(81.11)	17(18.89)
No	184	(67.2)	151(82.06)	33(17.94)

* Others: Other socio-health professions not included in the table.

**Table 3 healthcare-12-01230-t003:** Descriptive statistics of attitudes, self-confidence, and knowledge.

Dimensions n = 274	M	(SD)
**Attitudes**	**20.94**	**(2.86)**
I believe that health professionals should routinely evaluate bullying during childhood	3.57	(0.660)
I believe that childhood bullying is a major health problem.	3.76	(0.550)
I believe that childhood bullying is a public health problem and needs more attention and interventions.	3.70	(0.610)
I believe that some modalities/behaviors within childhood bullying are part of the growth/development of children.	2.56	(1.151)
I think adults should intervene when they suspect a child is being bullied.	3.83	(0.480)
I believe that primary care health professionals have an important role in helping to reduce childhood bullying.	3.53	(0.712)
**Self-confidence**	**18.03**	**(4.915)**
I am confident that I can recognize the signs and symptoms of bullying and victimization.	2.49	(0.691)
I would know how to act and proceed if a boy or girl tells me that he or she is being harassed or has been harassed.	2.30	(0.763)
I am confident in my ability and knowledge to screen my patients for bullying.	2.23	(0.724)
I am confident that I can intervene effectively with my patients who are harassed.	2.24	(0.761)
I have the skills to counsel patients who are being harassed	2.22	(0.742)
I know what to do if children tell me that they bully other people.	2.24	(0.733)
I am confident that I can intervene effectively with my patients who are harassing other people.	2.16	(0.722)
I have the skills to counsel patients who are harassing other people.	2.16	(0.712)
**Knowledge**	**51.92**	**(6.70)**
Bullying is considered verbal, physical or psychologically aggressive behavior.	3.65	(0.522)
For a child to be bullied, there must be a perceived relationship with an imbalance of power between the bully and the victim	3.05	(0.830)
The younger a child is, the more likely they are to tell an adult about bullying behavior.	2.50	(0.835)
In order to consider a boy or girl a victim of bullying, the bullying behavior must have the characteristic of intentionality on the part of the aggressor.	2.62	(0.870)
One of the characteristics of children who are victims of bullying is that they tend to be insecure.	2.87	(0.831)
Children who are victims of bullying often have difficulty sleeping.	3.39	(0.551)
Girls are more likely to use less overt behaviors, such as psychological manipulation, when bullying, compared to boys.	3.05	(0.749)
Children who are perceived as “different” are more at risk of being bullied	3.55	(0.579)
Compared to girls, boys are more likely to physically and verbally bully other people.	2.92	(0.806)
Children who are victims of bullying often complain of abdominal pain and headaches.	3.41	(0.543)
Overweight children are more likely to be bullied.	3.48	(0.536)
Children who bully other people are more likely to be involved in violent roles in the future.	3.40	(0.579)
Children who are victims of bullying are at risk of depression and low self-esteem in the future.	3.56	(0.512)
Children who are exposed to violence at home are more likely to bully other people.	3.43	(0.578)
Children who are autistic, have ADHD, or who consider themselves to have a “different” sexual orientation or gender, are more likely to be bullied. *	3.56	(0.553)
According to the SEMFyC** the close and frequent relationship with children and their parents, have a privileged position to carry out early detection of bullying situations, in addition to the knowledge that the professional has. health of the child’s socio-familial context”	3.47	(0.680)

* ADHD: attention deficit hyperactivity disorder. **SEMFyC (Spanish Society of Family and Community Medicine).

**Table 4 healthcare-12-01230-t004:** Bivariate analysis between the dimensions of the questionnaire and the bullying screening carried out in consultation.

Bullying Screening						
	YES		NO			
n = 274	Me	RIQ	Me	RIQ	Mann–Whitney U	*p*
Attitudes	21.0	4	21.0	3	7468.0	0.208
Self-confidence	20.0	8	17.0	4	5430.5	<0.001
Knowledge	53.0	13	50	9.5	6359.0	0.002

**Table 5 healthcare-12-01230-t005:** Resources for addressing and preventing bullying.

Variables n = 274	n	(%)
**I have protocols and/or guides for the detection of bullying in my work**		
Yes	33	(12.0)
No	112	(40.9)
I’m not sure	129	(47.1)
**I have protocols and/or guides for the detection of bullying in my work**		
Yes	11	(4.0)
No	201	(73.4)
I’m not sure	62	(22.6)
**I believe primary health care professionals need additional training to address bullying.**		
Yes	30	(10.9)
No	182	(66.4)
I’m not sure	62	(22.6)

**Table 6 healthcare-12-01230-t006:** Aspects related to clinical practice linked to the approach and prevention of bullying.

Variables n = 90	n	(%)
**I intervene with my patients when I suspect that they are involved in a bullying problem:**		
Never	4	(4.4)
Rarely	16	(17.8)
Occasionally	27	(30.0)
Frequently	34	(37.8)
Very often	9	(10.0)
**I provide advice to the patient and family when a patient is being harassed or harasses other people.**		
Totally disagree	3	(3.3.)
Disagree	4	(4.4)
Agree	55	(61.1)
Totally agree	28	(31.1)
**I refer patients to a mental health professional when they are being harassed or harassing others.**		
Totally disagree	7	(7.8)
Disagree	4	(4.4)
Agree	40	(44.4)
Totally agree	39	(43.3)
**I contact the school counselor when a patient is being harassed or harasses others.**		
Totally disagree	15	(16.7)
Disagree	36	(40.0)
Agree	26	(28.9)
Totally agree	13	(14.4)
**I provide some type of supporting material or information.**		
Totally disagree	26	(28.9)
Disagree	32	(35.6)
Agree	23	(25.6)
Totally agree	9	(10.0)
**Document/record in the patient’s medical history when he/she has been harassed or if he/she has harassed other people.**		
Totally disagree	5	(5.6)
Disagree	7	(7.8)
Agree	40	(44.4)
Totally agree	38	(42.2)

## Data Availability

The data are available upon email request to the corresponding authors.

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
