# Peer review of "Attitudes, Self-Confidence, and Knowledge of Primary Care Professionals towards School Bullying"

_healthcare, 2024, doi:10.3390/healthcare12121230_

Round 1
Reviewer 1 Report
Comments and Suggestions for Authors
Thank you for allowing me to review this paper. Addressing bullying from a multidisciplinary perspective is key to prevention and early intervention. This work focused on the attitudes, practices and knowledge of primary care professionals is also of interest.
Introduction: The phenomenon, prevalence and consequences it presents for the health and general well-being of the students involved are defined with appropriate appointments. However, what definition of bullying they take as a reference is missing, and the corresponding quote can be added (probably from Olweus). On the other hand, the paragraph that runs from line 68 to line 74 could be expanded to include more information about each of the study variables. Thus, there is a lot of information in the introduction about bullying but not much appears about the current state of research on the study variables (attitudes, practices and knowledge). It would be positive to expand this paragraph, explaining the results of other similar investigations or, at least, explaining other investigations that have worked with some of them. The most necessary information in the introduction is to explain to the reader the current state of research on the specific topic, which in the case of this article is attitudes, practices, knowledge and training needs of primary care professionals. Therefore, I recommend that this information be completed.
The research design is appropriate. Research hypotheses are not proposed; they could be included. The sample size is considerable, although only a third of the total participants who responded to the questionnaire responded to the open responses, which is recognized as a limitation of the study. There is a great difference in the number of responses between professional groups, which has been included as a limitation of the study. Regarding the study variables and the instrument used, it is well constructed and explained, being a development carried out by the same research team. The data analysis is appropriate according to the nature of the study variables. The ethical considerations they present are appreciated.
The results are adequate in general terms. It is necessary to review the structure presented in Table 2 to leave a first row where n and % are specified, where gender does not appear. Gender should appear in the bottom row. A left column should be created to include the variables on the left and the categories in the next column to the right. If you prefer not to create a new column, you can bold the variables. This will make the information easier to understand. With respect to Table 4, only the title is found. Please include the table below the title.
The discussion is correct. If study hypotheses are included, they could be contrasted in this section. The studies cited in this section could be explained in the introduction to explain the current state of research on the study variables.
The limitations are realistic with respect to the research carried out. The conclusions summarize the results of the research successfully.
We hope that the suggested recommendations are useful for improving the article.
Author Response
Dear Reviewer:
We sincerely appreciate your constructive comments on our article. We have taken each of your points into consideration and have made the following modifications. Please find a detailed description below of how we have responded to each of them. We have also attached a revised manuscript, where changes are highlighted in yellow.
Comment 1. Introduction: The phenomenon, prevalence and consequences it presents for the health and general well-being of the students involved are defined with appropriate appointments. However, what definition of bullying they take as a reference is missing, and the corresponding quote can be added (probably from Olweus).
- Response: I have completed the definition of bullying and added the reference accordingly. You can verify the changes between lines 29 and 34.
Comment 2. On the other hand, the paragraph that runs from line 68 to line 74 could be expanded to include more information about each of the study variables. Thus, there is a lot of information in the introduction about bullying but not much appears about the current state of research on the study variables (attitudes, practices and knowledge). It would be positive to expand this paragraph, explaining the results of other similar investigations or, at least, explaining other investigations that have worked with some of them.
- Response: I have made the modifications to the text as suggested, including additional information. I have added the information between lines 83 and 94.
Comment 3. The most necessary information in the introduction is to explain to the reader the current state of research on the specific topic, which in the case of this article is attitudes, practices, knowledge and training needs of primary care professionals. Therefore, I recommend that this information be completed.
- Response: Added information between lines 83 – 94.
Comment 4. The research design is appropriate. Research hypotheses are not proposed; they could be included. The sample size is considerable, although only a third of the total participants who responded to the questionnaire responded to the open responses, which is recognized as a limitation of the study. There is a great difference in the number of responses between professional groups, which has been included as a limitation of the study. Regarding the study variables and the instrument used, it is well constructed and explained, being a development carried out by the same research team. The data analysis is appropriate according to the nature of the study variables. The ethical considerations they present are appreciated.
- Response: Thank you for the suggestion. We include the study hypothesis between lines 100-102.
Comment 5. The results are adequate in general terms. It is necessary to review the structure presented in Table 2 to leave a first row where n and % are specified, where gender does not appear. Gender should appear in the bottom row. A left column should be created to include the variables on the left and the categories in the next column to the right. If you prefer not to create a new column, you can bold the variables. This will make the information easier to understand.
- Response: The columns for gender with n and (%) have been added to Table 2 as requested.
Comment 6. With respect to Table 4, only the title is found. Please include the table below the title.
- Response: Done
Comment 7. The discussion is correct. If study hypotheses are included, they could be contrasted in this section. The studies cited in this section could be explained in the introduction to explain the current state of research on the study variables.
- Response: The information has been added between lines 317-321
Reviewer 2 Report
Comments and Suggestions for Authors
The work presented is of enormous interest and relevance in providing valuable information on the practices and perceptions of primary health professionals regarding bullying. With some improvements in structure and detail, it could become an even more useful resource for the scientific and professional community.
Below are a number of aspects that could improve the work presented:
1. The abstract lacks information on sample size, the specific context of the study, and key findings that could be more salient.
2. The introduction is adequate and provides good context, but it would be beneficial to include more recent statistics on the prevalence of bullying globally and locally. This would help to better place the study in a broader context.
3. Although the references are relevant, it would be useful to include more recent studies that address successful interventions in the management of bullying by primary health professionals.
4. The mixed design is adequate, but the description of the qualitative phase could benefit from further elaboration on the coding process and thematic analysis.
5. The methods section is well described, but it would be helpful to explain why the nonprobability sampling approach was chosen and how they ensured that the sample was representative.
6. The conclusions are well supported by the results, but it would be useful to include more concrete practical recommendations for health professionals.
Author Response
Reviewers 2's comments:
Dear Reviewer:
We sincerely appreciate your constructive comments on our article. We have taken each of your points into consideration and have made the following modifications:
We have considered your suggestions and made the following changes:
Comment 1. The abstract lacks information on sample size, the specific context of the study, and key findings that could be more salient.
- Response: The text has been modified between lines 12 a 25.
Comment 2. The introduction is adequate and provides good context, but it would be beneficial to include more recent statistics on the prevalence of bullying globally and locally. This would help to better place the study in a broader context.
- Response: I have added the information between lines 49-54.
Comment 3. Although the references are relevant, it would be useful to include more recent studies that address successful interventions in the management of bullying by primary health professionals.
- Response: I have added the information between lines 83 and 94.
Comment 4. The mixed design is adequate, but the description of the qualitative phase could benefit from further elaboration on the coding process and thematic analysis.
- Response: I have added more information between lines 175 -187.
Comment 5. The methods section is well described, but it would be helpful to explain why the nonprobability sampling approach was chosen and how they ensured that the sample was representative.
- Response: Thank you for your comment. The sample was specifically chosen to include primary care professionals who are more likely to be in contact with populations affected by school bullying, such as community, pediatric, and school nurses, as well as general practitioners and pediatricians, including both medical and nursing residents from these specialties. A non-probabilistic sampling method was used due to its practicality and feasibility in accessing a specific group of healthcare professionals who have direct relevance to the study's focus.
To ensure the sample was as representative as possible within these constraints, we aimed to include a diverse range of professionals across different roles and levels of experience within primary care settings. The final sample consisted of 274 participants, of which 94 responded to the open-ended questions constituting the qualitative phase of the study. This approach enabled us to gather in-depth insights from those directly involved in the care of populations affected by school bullying. The characteristics of the sample, detailed in the results section, indicate a comprehensive inclusion of relevant professionals, enhancing the reliability and relevance of our findings.
Comment 6. The conclusions are well supported by the results, but it would be useful to include more concrete practical recommendations for health professionals.
- Response: The information was added to the text between lines 402 y 410.
Reviewer 3 Report
Comments and Suggestions for Authors
The manuscript examined the attitudes, practices, and knowledge of primary care professionals regarding school bullying. The authors used mixed methods to collect the data. They found a positive correlation between higher self-confidence and knowledge. Also, the results indicate low overall self-confidence in addressing bullying. The topic that the researchers tackled is significant for advancing knowledge on bullying. However, there are several problems with the work.
- The abstract is not clear enough; hence the need to rewrite it.
- The entire work needs editing with respect to grammar
- In the analysis, the authors mentioned table 4 on page 7. Yet, that table is not found in the work.
- In the discussion section, the authors mentioned that "This highlights an exciting opportunity for further research and development in the field, aiming to bridge the knowledge gap among primary care doctors and nurses regarding bullying, which is crucial for its effective management." Yet, the authors never mentioned this in the literature review section.
The quality of English in the abstract is not good.
Author Response
Reviewers 3's comments:
Dear Reviewer:
We sincerely appreciate your constructive comments on our article. We have taken each of your points into consideration and have made the following modifications:
We have considered your suggestions and made the following changes:
Comment 1. The abstract is not clear enough; hence the need to rewrite it.
- Response: The text was modified between lines 12 a 25.
Comment 2. The entire work needs editing with respect to gramar
- Response: A comprehensive analysis of the text was carried out by a native English speaker.
Comment 3. In the analysis, the authors mentioned table 4 on page 7. Yet, that table is not found in the work.
- Response: Done
Comment 4. In the discussion section, the authors mentioned that "This highlights an exciting opportunity for further research and development in the field, aiming to bridge the knowledge gap among primary care doctors and nurses regarding bullying, which is crucial for its effective management." Yet, the authors never mentioned this in the literature review section.
- Response: This issue is addressed in the literature review section, specifically in lines 92 to 95.
Round 2
Reviewer 2 Report
Comments and Suggestions for Authors
The review performed is adequate to the requirements made by this reviewer.